# Reliability and Practical Use of a Commercial Device for Measuring Punch and Kick Impact Kinetics

**DOI:** 10.3390/sports10120206

**Published:** 2022-12-13

**Authors:** Luke Del Vecchio, John Whitting, Jennifer Hollier, Annabelle Keene, Mike Climstein

**Affiliations:** 1Faculty of Health, Southern Cross University, Bilinga, QLD 4225, Australia; 2Physical Activity, Sport and Exercise Research (PASER), Faculty of Health, Southern Cross University, Bilinga, QLD 4225, Australia; 3Exercise and Sport Science Exercise, Health & Performance Faculty Research Group, Faculty of Health Sciences, University of Sydney, Sydney, NSW 2000, Australia

**Keywords:** punching, kicking, peak force, combat athlete, impact kinetics

## Abstract

Martial arts, boxing and combat sports such as mixed martial arts participation have gained popularity in recent years internationally. One common aspect to these sports is the training and skill in maximizing strike impact of punches and kicks, referred to as impact kinetics, with commercial devices now available to assess punching and kicking power in athletes training facilities and gyms. We, therefore, assessed the reliability of a commercial device, the PowerKube^TM^ (Strike Research Ltd., Norwich, England) via the technical error of measurement (TEM) in both linear and non-linear simulated strikes to the center of target, off-center, level and inclined in a laboratory setting. The highest mean impact power resulted from level, center strikes (5782 ± 230 W) followed by level, off-center strikes (4864 ± 119 W, *p* < 0.05), inclined center strikes (4500 ± 220 W, *p* < 0.05), and inclined, off-center strikes (3390 ± 151). Peak power reductions ranged from 15.9% (level, off-center) to a maximum of 41.4% (incline, off-center) compared to the level, center strikes. Coaches are advised to take steps such as videoing strikes with high sampling rates to better ensure consistency in impact orientation, being perpendicular and centered on the strike pad, to best capture the peak power of kicks or punches.

## 1. Introduction

Interest and participation in self-defense, martial arts, boxing and combat sports have increased internationally over the last decade internationally [1,2]. The emphasis of these combat activities is primarily focused upon maximizing impact concerning strikes by either the hand (punches) or feet (kicks) to inflict maximal force to either physically stop or hurt the opponent or attacker or to score points in a competition [3].

In trained individuals, these strikes, for defense or points during competition, are powerful and highly effective and not surprisingly, associated with injury. For example, in mixed martial arts, over 78 of 220 (35.5%) competitors in an ultimate fighting championship experienced 96 injuries [4]. In another study conducted by Kazemi and Pieter [5], involving 318 Taekwondo athletes competing at national championships, 13 percent experienced injuries. In male and female participants, most injuries were to the lower extremities and consisted of sprains, joint dysfunction, contusions and lacerations. On occasion, deaths have been reported in mixed martial arts competitions, particularly from brain injuries resulting from direct blows to the head [6,7].

Punching and kicking power are crucial for self-defense, boxing, mixed martial arts (MMA), combat sports, and martial arts, particularly where full contact is involved. The measurement of kicking and punching strikes power is generally referred to as impact kinetics and has been previously investigated in various styles of martial arts, including karate, Korean Taekwondo [8,9,10,11], boxing [12,13] and more recently in combat athletes participating in mixed martial arts (MMA) [14,15]. Lenetsky and colleagues [16] recently completed a review of different techniques to assess strike force. They reported wide variability in assessing punching and kicking power by either inertial devices, via direct and indirect force measurements or taken directly from the striking limb by athlete-worn devices. Impact force is commonly assessed in the literature in combat athletes and has been measured by force plates [17,18,19], dynamometers [20], dynamometric bags with embedded accelerometers [14], striking bags with embedded strain gauges [21], ballistic pendulums [22] and strain gauges [8,23].

Using a padded force plate, Gulledge et al. [11] investigated peak force and impulse in the reverse and three-inch power punch. They found the force of the reverse punch was almost twice that of the power punch (790 vs. 1450 N, *p* < 0.001). Gavagan et al. [24] investigated the round house kick in advanced blackbelts in Muay Thai, Karate and Taekwondo using a strain gauge attached to a pad. They reported peak forces of 1211 N (Karate), 1400 N (Muay Thai) to 1547 N (Taekwondo). Similarly, Busko and Nikolaidis [8] compared the impact forces of Taekwondo athletes completing punches and kicks in a laboratory using a punching bag with a built-in strain gauge. Using similar technology, they reported substantially higher peak forces than Gavagan et al. [24] in both punches (straight 1659 N, hook 1843 N) and kicks (rear kick 3541 N). Although not necessarily explaining the variability in results demonstrated here, technique and striking context also influence the magnitude of peak forces. For instance, Busko and Nikolaidis [8] concluded that the kicking values in the simulated fight in their study were lower than the forces measured in individual kicks.

In an attempt to standardize these measurements, a commercial device, the PowerKube^TM^, (Strike Research Ltd., Norwich, England) has been developed to assess impact power in martial artists, boxers and combat athletes. According to its manufacturer, measurements with such devices are sensitive to strike area and materials. This device consists of a proprietary construction of plates, foams, pressure sensors and accelerometers integrated with performance mapping software to provide instantaneous outputs of kicking or punching power and energy via a USB-connected laptop. At present, there is limited scientific literature reporting on the use of the StrikeMate^TM^(Strike Research Ltd., Norwich, England) to measure striking power (absolute or relative) production in combat athletes. Del Vecchio and colleagues [15] used the StrikeMate^TM^ to investigate the effects of six-weeks of power weight training on peak punching and kicking power. They reported increases in peak power of 7.2 percent in the front kick and 34.0 percent in the roundhouse kick following the six-weeks of power training. However, a close investigation of the mean changes, specifically the standard deviation (+ SD) reported in this study, identified a high variability in peak power, ranging from 25.7 percent in the front kick to 40.0 percent in the jab punch.

Galpin and colleagues [25] investigated the reliability of the StrikeMate^TM^ using three kettlebell weights (8.0, 17.9 and 24.0 kg) dropped linearly to the center of the device from a height of 100 cm. Their methodology involved dropping the kettlebell weights directly onto the StrikeMate^TM^. However, this methodology is flawed as it assumed the kettlebell would land on the StrikeMate^TM^ in a consistent orientation each time, potentially significantly influencing the impact force and, therefore, the power output measured. Another limitation of the study by Galpin et al. was that it only attempted to measure impact power for linear strikes in the center of the StrikeMate^TM^. Galpin et al. [25] reported a coefficient of variance (CV) ranging from 4.52% (17.9 kg drop weight) to 6.01% (8.0 kg drop weight). A new commercial model has superseded the StrikeMate^TM^ by the same manufacturer, the PowerKube^TM^, which the authors of the current study have pilot-tested with highly experienced, competitive combat athletes. Anecdotally, however, we observed high variability in the power output when athletes were punching and kicking the device, despite their experience as combat athletes and martial artists. When their strikes (punches and kicks) appeared to be non-linear (e.g., hook punch or roundhouse kick) and contacted the pad at an angle that was not perpendicular, and/or impacted the pad slightly off-center, this variability in power output appeared to be more prevalent. Therefore, given the variability seen in the Del Vecchio and colleagues study [15] with the original StrikeMate^TM^ and our anecdotal observations of the updated PowerKube^TM^, the primary purpose of the current study was to perform a highly systematic investigation to determine the reliability of simulated impact strikes to the PowerKube^TM^. Therefore, we hypothesize that there will be a significant difference in impact force when simulated strikes were delivered under the different conditions (perpendicular versus at an angle; center versus off-center) and a combination of off center and angled).

## 2. Materials and Methods

This study was an in vitro laboratory based study whereby no participants were involved in the study, rather we engineered a drop system to assess impact forces. An in vitro method was utilized as in vitro is recognized for having high standards of reproducibility and reliability [26].

To determine the reliability of the PowerKube^TM^, we developed a fixed, steel construction, vertical support system which was permanently mounted to a concrete slab to suspend a weighted mass over both a force platform (Kistler Type 9287, Kistler Instrumente, AG, Winterthur, Switzerland) and a PowerKube^TM^ (see Figure 1). Following pilot testing, a firm, rounded mass (weighted medicine ball—8.04 kg) (Buffalo Sports, Victoria, Australia) was selected as the impact mass as it would most likely result in high repeatability concerning impact angle given its spherical shape (Figure 1 and Figure 2). The mass was suspended above the force platform and PowerKube^TM^ using a customized sling and release system that allowed for high release consistency (negligible horizontal movement). A commercial plumb bob was used to ensure the mass drops from the sling support were either dead center on the PowerKube^TM^ or 40 mm off-center. The plumb bob was used prior to each condition and intermittently during conditions to ensure impacts were consistently delivered to two specific locations on the PowerKube^TM^, dead center or 40 mm off-center.

In order to determine the reliability and technical error of our drop method prior to testing with the PowerKube^TM^, we first performed 40 drops directly onto the force platform. This consisted of two sets of 20 vertical drops of the 8.4 kg cylindrical mass vertically onto the force plate. We assessed the coefficient of variation (CoV) and technical error of measurement (TEM) from the data collected.

Given the variability observed previously in strike locations and angles relative to the target center and orientation, the reliability and technical error of simulated strikes to the PowerKube^TM^ were assessed under four specific conditions, with 40 simulated strikes per condition. Condition order was randomized; however, all 40 strikes were completed before changing conditions.

Condition 1 (LC) involved leveled PowerKube^TM^ and centered drops: It simulated a linear strike with a perpendicular drop to the center of the PowerKube^TM^. Condition 2 (LO) involved a leveled PowerKube^TM^ and off-center drops: It simulated a linear strike with a perpendicular drop to the PowerKube^TM^, offset from center by 40 mm. Condition 3 (IC) involved an inclined PowerKube^TM^ and centered drops: It simulated non-linear strikes applied to the center of the PowerKube^TM^ with the device angled 12 degrees from perpendicular. Condition 4 (IO) involved an inclined PowerKube^TM^ and off-center drops: It also simulated non-linear strikes with the PowerKube^TM^ angled at 12 degrees from perpendicular; however, this time with the drop delivered 40 mm offset of center.

Raw data were sampled from the force plate at 10,000 Hz using Nexus data collection software (Nexus 2, Oxford, UK) and were used unfiltered. Pilot testing with filtering revealed negligible differences in absolute values and no differences in statistical results. Data from the PowerKube^TM^ were determined directly from the proprietary software (PowerKube, September 2020). For each strike, The PowerKube^TM^ software provided four different peak strike values: impact power (W); impact power (ft.lbs/s); kinetic energy (cal × 10) and the power index (no units). The power index and kinetic energy outcome measures are unitless, proprietary calculations.

### 2.1. Simulated Strikes

One investigator stood motionless alongside the suspended weighted mass and ensured no horizontal or vertical movement to the suspended mass. When the mass was determined to be motionless, the investigator removed the supporting hand and then used the trigger mechanism to release the mass from suspension and allow it to drop and strike the PowerKube^TM^. The timer was triggered on the PowerKube^TM^ prior to each simulated strike, in order to initiate data collection. The power output summary displayed from the software onto the device was entered manually into Excel.

### 2.2. Statistical Analysis

Data were initially entered into Excel and then transferred to SPSS for subsequent analyses (Version 27.0; IBM Company, Armonk, NY, USA). Descriptive statistics, technical error of measurement (TEM), relative TEM (%TEM) [26] and the coefficient of variance (CoV) were used to assess the reliability of the drop method of the PowerKube^TM^ [27]. A bivariate Pearson correlation with a two-tailed test of significance was conducted on selected outcome variables to determine the strength of any relationships. The 95% confidence intervals (95% CI) A one-way analysis of variance with a Bonferroni post hoc tests was used to determine significance between mean drops for reliability of the drop method. Significance level was set a priori at *p* < 0.05.

The level of agreement (LoA) between the related quantitative measurements is illustrated through Bland–Altman plots [27], with the corresponding 95% LoA, using the formula: mean difference between measures ± SD. Standard error of the mean (SEM) was calculated as dividing the standard deviation (SD) by the square root of the number of values in the data set (√) [28]. We also calculated the mean absolute percentage error (MAPE). Mean absolute percentage error (MAPE) values were calculated as the average test 1 value of the errors of each device relative to the test 2 measures [29].

## 3. Results

### 3.1. Drop Method Coefficient of Variation and Technical Error of Measurement

The CoV and CoV% for each set of 20 drops are presented in Table 1 below. The absolute TEM and the relative TEM% between these sets are shown in Table 1.

### 3.2. PowerKube^TM^ Reliability

The outcome variables from the PowerKube^TM^ from all four conditions are presented in Table 2. With regard to peak power, the highest mean value was found in Condition 1 with the PowerKube^TM^ level and the strike to center. Conditions 2, 3 and 4 resulted in peak powers that were 84.1%, 77.8% and 58.6% of peak power in Condition 1, respectively.

Similar declines in output were observed in the other two power measures across conditions 2 to 4 compared with condition 1, with outputs ranging from 58.6% to 84.1% of condition 1 (Table 2). However, concerning the kinetic energy measurement, we found a negligible increase (+1.0%) in condition 2, compared with condition 1. When the PowerKube^TM^ was inclined in conditions 3 (−23.4%) and 4 (−33.1%) there was again a significant decrease (*p* < 0.05) in kinetic energy compared with condition 1.

There were significant relationships for Condition 1 (level center) in impact power (W) and impact power (ft/lbs/s) (*p* < 0.001, r = 0.997, 95% CI 0.994–0.998). There were also significant relationships in condition 1 impact power to the power index (*p* < 0.001, r = 0.993, 95% CI 0.987–0.996) and to kinetic energy (*p* < 0.001, r = 0.525, 95% CI 0.255–0.719).

With regard to test 1 and re-test 2, correlations between conditions were all significant (*p* < 0.001) and ranged from r = 0.996 (flat, off-center) to r = 0.999 (incline, off-center).

Concerning the repeatability of each drop within each condition, a comparison of the first set of 20 drops to the second set of 20 drops per condition revealed relative TEM (TEM%) (an accuracy index) values ranging from 0.8% to 3.6% (see Table 3).

### 3.3. Root Mean Square Difference (RMSD)

The root mean square difference (RMSD) was calculated to assess the difference in the values between test 1 and the re-test 2. As illustrated in Table 2, the mean of all four conditions and the RMSD were near identical.

### 3.4. Limits of Agreement LoA), Standard Error of the Mean (SEM) and Mean Absolute Percentage Error (MAPE)

The mean absolute percentage error (MAPE) plot (Figure 2) identified the first three conditions. The MAPEs were very similar, ranging from 12.4% to 12.5%; however, the fourth condition was lower at 9.4%. Bland–Altman plots were also produced, which included the 95% LoA between tests (first 40 drops versus second 40 drops) for all four conditions (Figure 2, Figure 3, Figure 4 and Figure 5). The mean impact with center flat strikes (Figure 3) was 697 Watts, with the upper and lower LoA being 779.4 and 616.0 Watts, respectively (SEM 36.4 and 30.3, respectively, MAPE 12.5%). The mean impact with off-center flat strikes (Figure 4) lower at 591 Watts, with the upper and lower LoA being 637 and 544 Watts, respectively (SEM 18.8 and 15.4, respectively, MAPE 12.4%). When the PowerKube^TM^ was set at an incline, the mean center impact was 452 Watts, with the upper LoA at 526 Watts and the lower LoA at 378 Watts (Figure 5) (SEM 34.7 and 29.0, respectively, MAPE 12.4%). When the PowerKube^TM^ was at an incline and off-center impacts, the mean impact was 318 Watts with the upper and lower LoAs at 357 and 279 Watts, respectively (Figure 6) (SEM 23.8 and 20.7, respectively, MAPE 9.4%).

In the first three conditions, the Bland–Altman plots revealed only three data points were outside of the 95% LoA (3.3%) indicating very few outliers and very good agreement between our test and re-test for the first three conditions. In the fourth condition (incline, off-center) we found that 15% of the test drops fell outside of the lower LoA, indicating a slightly lower agreement between test and re-test.

## 4. Discussion

The main purpose of the current study was to perform a highly systematic investigation to determine the reliability of simulated impact strikes to the PowerKube^TM^. A secondary purpose was to determine whether there were significant differences in peak powers between strikes delivered under different conditions: (i) perpendicular versus at an angle to the pad, and (ii) center versus off-center on the target pad. Our hypothesis was confirmed since the highest mean impact power resulted from level, centered strikes, followed by level, off-center strikes; inclined center strikes; and inclined, off-center strikes.

Initially, the reliability of our drop method was determined to assess if our methodology was precise, with low variability. Since two sets of 20 drops with the spherical mass resulted in a very low CoV% (0.47) and a very low relative TEM% (0.350), our method of dropping the mass indicated a high level of precision and repeatability. These findings clearly justified our drop method, which was used for the subsequent assessment of the reliability of the PowerKube^TM^.

The reliability of the predecessor of the PowerKube^TM^, the StrikeMate^TM^ was previously evaluated by the Australian Institute of Sport [30]. Of particular interest was their assessment of the device’s typical error (TE) under two conditions. Linear simulated strikes by an impact pendulum, delivered to the center of the PowerKube^TM^ and 50 mm off-center. The study reported that the TE was only 0.9% (90% CI 0.6 to 2.7%) with center strikes and 1.1% (90% CI 0.7 to 3.4%) with the 50 mm off-center strikes. In the current study, however, with a more rigorous and systematic approach involving a higher number of drops delivered with a known level of accuracy, we reported relative TEM’s ranging from 0.8% to 3.6%. With only 0.35% of the error attributable to our method, the vast majority of the TEM’s displayed by our four conditions on the StrikeKube^TM^, are, therefore, due to error in the PowerKube^TM^. This should be considered when using the PowerKube^TM^ to assess differences in punching and kicking values.

Perhaps not unexpectedly, the inclined conditions (3 and 4) where the drop vector was at an angle of 12 degrees relative to perpendicular, displayed the highest relative TEM’s. Aside from the kinetic energy measure in condition 4, all power measures in conditions 3 and 4 had relative TEM’s of 3.2% and 3.6%. When struck in the center and at a vector perpendicular to the pad’s face (condition 1), the PowerKube^TM^ displayed relative TEM’s that were higher than in condition 2, when the drops were offset from the center by 40 mm. These results were unexpected, since the sensors in the pad are aligned with the center of the target. These results demonstrate that different strike vectors, delivered at different locations on the target, will result in varying levels of systematic error. Consequently, any analysis of punching or kicking power changes or differences with the PowerKube^TM^, must consider the relatively unpredictable error and should look for differences or changes that substantially exceed 3.6%.

The systematic error revealed by the TEM’s from our four conditions may not be the biggest concern, however, when planning punching and kicking power analyses with the PowerKube^TM^. In the current study, conditions 2 to 4 delivered a range of simulated strikes that were not dead center and perpendicular to the target. There were significant reductions in all three power measures in all three of these conditions compared with condition 1. Therefore, any strike to the PowerKube^TM^ that is offset from the dead center, delivered in a non-linear fashion or at an angle to the target, will significantly reduce power output. These reductions range from 15.9% to 41.4%, which far exceed the individual TEM’s for each of our four conditions. These differences in output, primarily dependent on the angle and location of the strike, may limit the device in making accurate and systematic scientific evaluations of performance differences or changes in certain circumstances. To use this device effectively for such testing, one must ensure consistency of strike angle and location on the pad, to approach the TEM’s measured in our study of up to 3.6%. While not insurmountable with appropriate technology and methodology, these are technical and methodological hurdles that must be considered.

### Practical Utilization

Given our laboratory findings from the drop tests and the identified decrease identified in peak power ranging from −15.9% (level, off-center) to −41.4% (incline, off-center) and to investigate the ecological validity of the PowerKube^TM^ device, we utilized existing data from professional combat athletes in our laboratory for evaluation of practical impact kinetic testing. All kicks and punches had been videoed with a smart phone (4 K at 60 frames per second). All athletes were currently training and competing in their respective combat sports.

The first athlete was a female professional boxer who completed a punching endurance protocol which consisted of 20 linear punches (rear hand punch) and 20 non-linear punches (hook). The mean of the 20-rear hand (linear) punches was 6835 watts (±1481). However, two of the strikes were off target (Figure 7) with lower impact values of only 4356 and 3978 watts.

As the two punches were off-center, we deleted them from the data set and reanalyzed the data. We found the true mean impact of the rear hand punches was higher (+4.3%) at 7132 watts (±1231). Although this is a non-significant difference (*p* = 0.255), at the elite level, this variation from true performance is critical to evaluating the effectiveness of strength and conditioning programs, particularly when evaluating peak and mean punching power.

We completed a similar evaluation with a male professional boxer. We found the mean of 20 non-linear strikes (hook punch) to be 11,973 watts. However, three of the strikes were off center and resulted in lower impact scores. When these off-center strikes were eliminated from the data set, this resulted in a higher (+8.7%, *p* = 0.089) mean impact for the 20 punches.

Athlete 3 was a professional kickboxer who completed a peak impact power kicking protocol of 4 sets of 5 repetitions (5RM), with one-minute rest between sets. Similar to the laboratory testing, we had the athlete complete linear and non-linear strikes on the PowerKube^TM^ Peak impact power was determined using a 5RM front kick (linear strike, lead leg). The athlete’s peak impact was 3801 watts; however, the other four kicks were off-center and low (Figure 8), confirmed via video, which resulted in much lower impact scores (−39.7% to −42.1%). We repeated the testing with a non-linear kick (rear leg, roundhouse) and athlete three had a peak impact of 48,533 watts. However, three kicks had much lower (range −8.52 to −9.34%) impact power (44,396, 44,290 and 44,000) and upon inspection of the video, these lower impact power kicks were seen to be off-center.

With respect to impact testing with combat athletes, we utilized a 5RM for both punches and kicks to determine the athletes’ maximal impact force. This five-strike technique, with short rest (3 s) between strikes as incorporated into the PowerKube^TM^ software for determining peak punch or kick force. The highest impact force was recorded.

When assessing muscular kick and/or punch endurance, we recommend the 20-strike technique with the mean of the 20 strikes representing the impact score. All strikes, 5 or 20-strike should be filmed for subsequent analysis, with any off-center strikes deleted from the data set.

## 5. Conclusions

In conclusion, the accuracy of our drop method should be accounted for when considering the accuracy of the PowerKube^TM^. These data suggest that less than one percent (0.35%) of the TEM calculated on the PowerKube^TM^ is likely attributed to the variability in our drop method. Given this consideration, we recommend that only linear strikes to center of the target pad are used for testing and monitoring peak punch and kick power in combat athletes. Ensuring consistency in strike vector orientation is therefore a necessary technological and methodological challenge of future investigators.

## 6. Strengths and Limitations

The strength to this study was our utilization of in vitro, laboratory-based testing of reliability of the PowerKube^TM^. Our drop method was shown to have a very low CoV (absolute and relative) and TEM (absolute and relative). Our drop testing removed the variability associated with human performance punching and kicking.

There were limitations to this study. Currently, there is no literature available on the StrikeKube^TM^, only the StrikeMate^TM^, thereby making comparisons difficult. Additionally, there is considerable variation in methodologies when assessing reliability of similar devices, again making comparison difficult.

## Figures and Tables

**Figure 1 sports-10-00206-f001:**
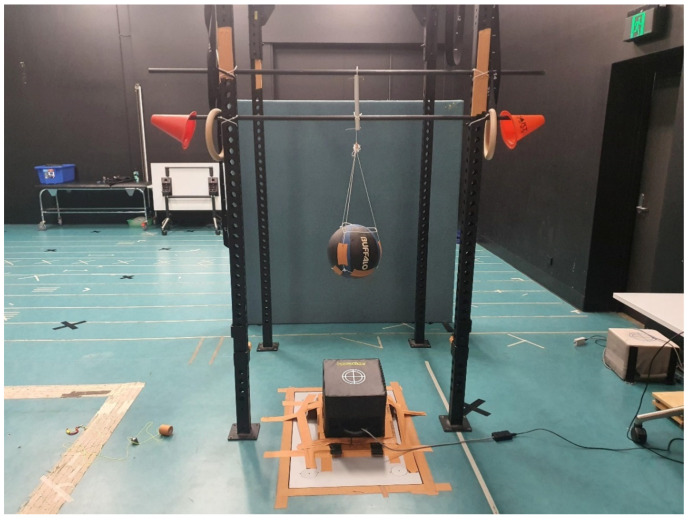
Weighted medicine ball suspended above the PowerKube^TM^ for assessing peak power.

**Figure 2 sports-10-00206-f002:**
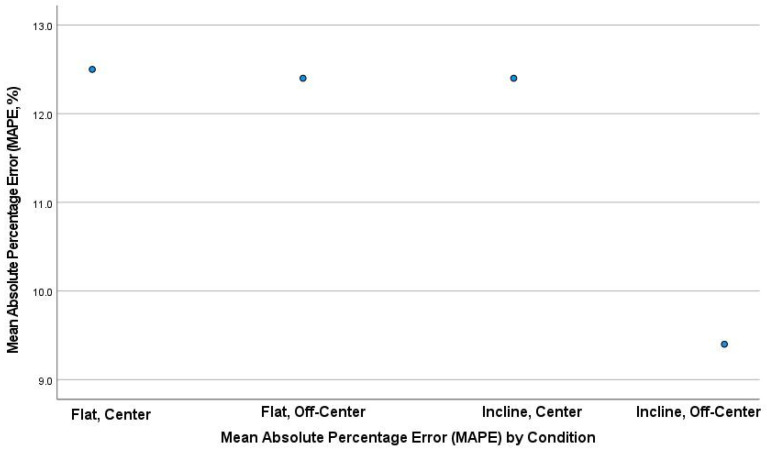
Mean Absolute Percentage Error (MAPE) for all conditions.

**Figure 3 sports-10-00206-f003:**
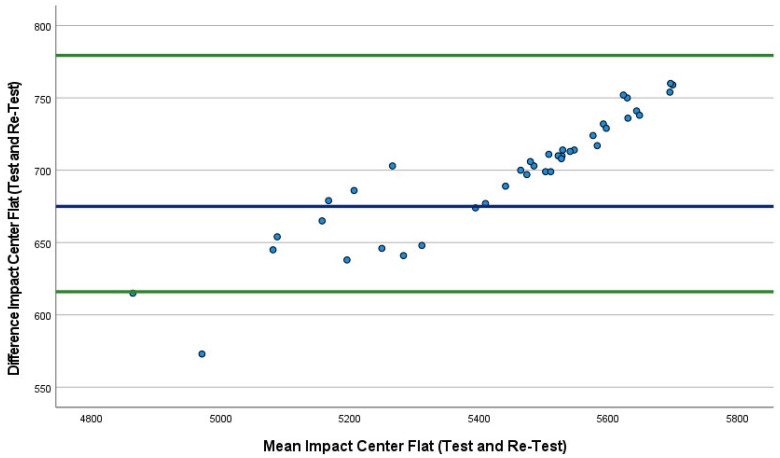
Bland–Altman plot indicating mean (blue line) and 95% limits of agreement (green lines) between flat center trial 1 (40 drops) and flat center trial 2 (40 drops).

**Figure 4 sports-10-00206-f004:**
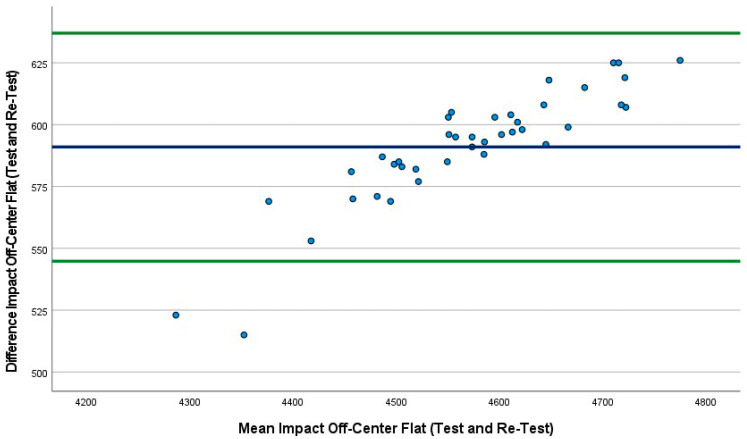
Bland–Altman plot indicating mean (blue line) and 95% limits of agreement (green lines) between flat off-center trial 1 (40 drops) and flat off-center trial 2 (40 drops).

**Figure 5 sports-10-00206-f005:**
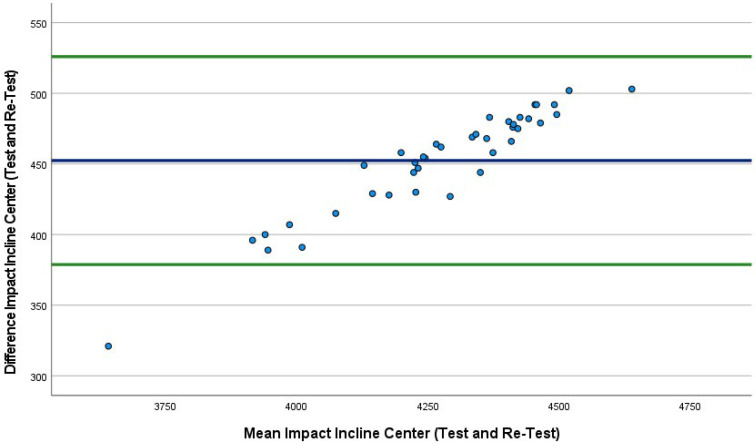
Bland–Altman plot indicating mean (blue line) and 95% limits of agreement (green lines) between incline center trial 1 (40 drops) and incline center trial 2 (40 drops).

**Figure 6 sports-10-00206-f006:**
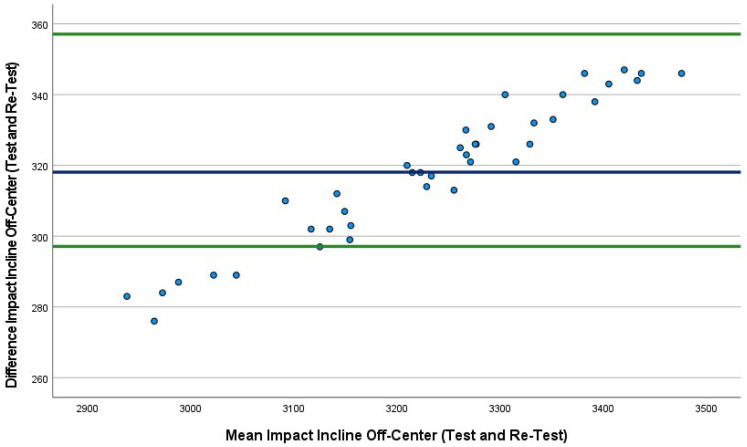
Bland–Altman plot indicating mean (blue line) and 95% limits of agreement (green lines) between incline off-center trial 1 (40 drops) and incline off-center trial 2 (40 drops).

**Figure 7 sports-10-00206-f007:**
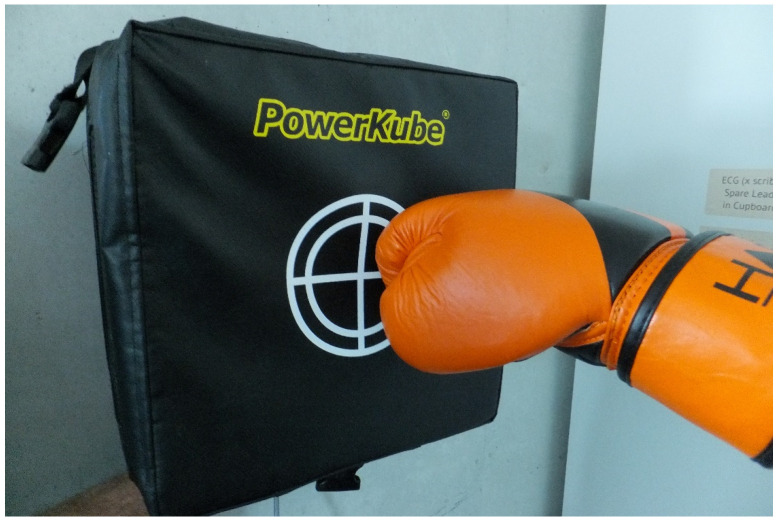
Boxer with linear strike off-center.

**Figure 8 sports-10-00206-f008:**
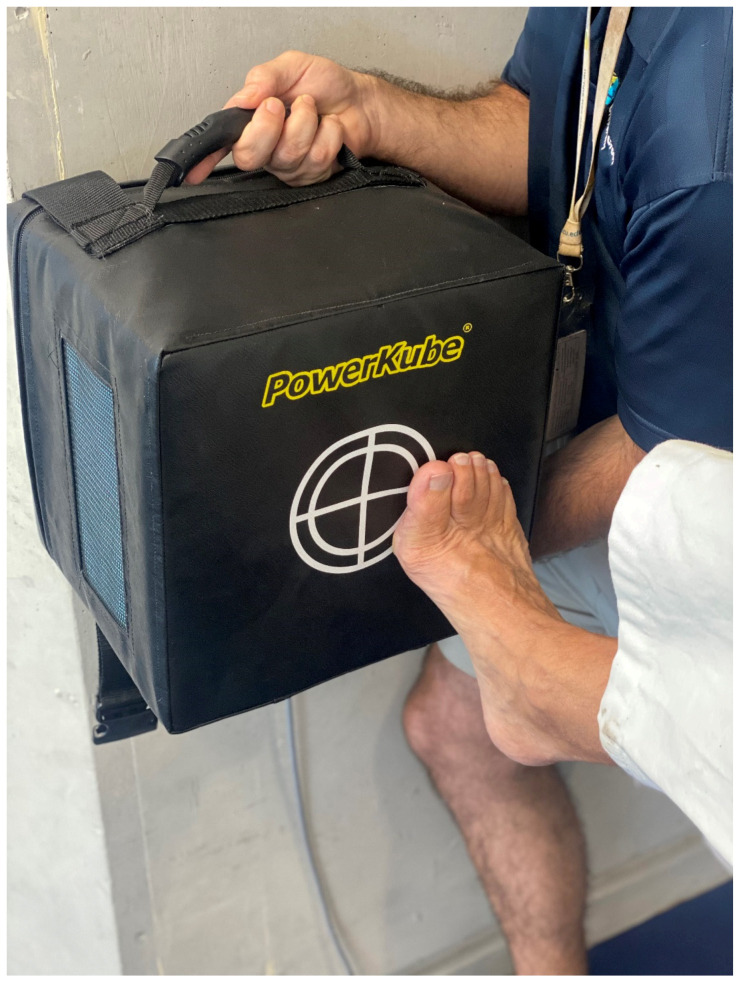
Mixed martial artist with linear kick off center.

**Table 1 sports-10-00206-t001:** Reliability and technical error of measurement of drop method.

	Set 1 (Drops 1–10)	Set 2 (Drops 11–20)
Force (N, mean ± SD) 95%CI (lower-upper)	4835.0 (22.9) 4824–4845	4863.6 (23.0) 4852–4874
CoV	0.00474	0.00472
CoV (%)	0.47	0.47
Absolute TEM between sets (N)	16.82	
Relative TEM (%) between sets	0.35	

Where: 95%CI = 95% confidence interval; CoV = coefficient of variation; TEM = technical error of measurement.

**Table 2 sports-10-00206-t002:** PowerKube^TM^ impact results. Values are mean ±SD, 95%CI (lower-upper), (Root mean square difference). (Where * = *p* < 0.05 from level center condition, LC).

Condition	Impact Power (W)	Impact Power (ft∙lbs/s)	Kinetic Energy (cal × 10)	Power Index
1 (LC)	5782 (230) (5707–5854) (5785)	5084 (192) (5022–5145) (5087)	47.5 (1.6) (46.9–48.0) 47.5	57.8 (2.3) (57.1–59.0) (57.9)
2 (LO)	4864 * (119) (4826–4902) (4865)	4273 * (98) (4241–4304) (4274)	47.9 * (1.1) (47.6–48.3) (47.9)	48.6 * (1.2) (48.2–49.0) (48.6)
3 (IC)	4500 * (220) (4430–4570) (4505)	4048 * (184) (3989–4106) (4052)	38.5 * (1.8) (38.0–39.1) 38.5	45.1 * (2.3) (44.3–45.8) (45.1)
4 (IO)	3390 * (151) (3341–3437) (3392)	3072 * (131) (3029–3114) (3074)	35.7 * (0.9) (35.4–36.1) (35.7)	33.9 * (1.5) (33.4–34.4) (33.9)

Where: LC = level center; LO = level off-center; IC = incline center; incline off-center.

**Table 3 sports-10-00206-t003:** PowerKube^TM^ technical error (absolute and relative).

Condition	Impact Power (W)	Impact Power (ft∙lbs/s)	Kinetic Energy (cal × 10)	Power Index
1 (LC)				
• TEM	174.9	146.4	1	1.8
• Tem%	3	2.9	2.1	3.1
2 (LO)				
• TEM	50.2	44.6	0.4	48.6
• Tem%	1	1	0.8	1.2
3 (IC)				
• TEM	156.1	129.7	1.3	1.6
• Tem%	3.5	3.2	3.3	3.6
4 (IO)				
• TEM	118.3	103.2	0.7	1.2
• Tem%	3.5	3.4	2	3.6

Where: TEM = technical error of measurement.

## Data Availability

Data supporting this study can be attained upon request by contacting the Principle Investigator.

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
