# Peer review of "Reliability and Practical Use of a Commercial Device for Measuring Punch and Kick Impact Kinetics"

_sports, 2022, doi:10.3390/sports10120206_

Round 1

Reviewer 1 Report

First of all, I congratulate the researchers for their efforts. This research is well designed but has some weaknesses. Performing the reliability measurements in two separate sessions could have revealed the reliability of the research in more detail.

-I think statistical analysis is not enough.

-In particular, reliability between sets and between conditions needs to be tested with ICC analysis. Both sets and conditions and the reliability of all tests should be presented to the readers. That is, both the internal reliability of two separate sets and four separate conditions, and the overall reliability of the sets and conditions.

-RMSE, MAPE, SWC and MDC analyzes should be performed in the reliability analysis.

-In order to better understand the reliability of the measurements by the readers, the limit of agreement values of the sets and conditions should be presented with the Bland-altman graph (lower and upper LOA values).

-Pictures in heading 4.1 break the article flow.

-You can not specify the pictures in a different place.

-Discussion and results should be revised after revisions.

***After revision, I would like to review the article again.

Best Regards.

Reviewer 2 Report

Abstract

Per the stated purpose (i.e., to assess the reliability of the PowerKubeTM via the TEM), please report the TEM in the abstract. As is, only impact power in different strikes is reported.

Introduction

Overall, this needs to be shortened with clear focus on your development of your problem / hypothesis. The focus needs to be on the “reliability” issue with secondary focus on the impact power. As is, the introduction jumps around lots and lacks clear focus.

Line 28 – Internationally is repeated twice here; please fix.

Lines 32-40 (the 2nd para) – Please delete as this discussion of injuries has nothing to do with the stated purpose and does not help develop your problem / hypothesis.

Line 47 – The reader would benefit from knowing the range of “wide variability” (i.e., x to y).

Line 54-67 – Suggest deleting this paragraph as it does not develop your problem / hypothesis and as is, interrupts the flow related to reliability.

In line 69, you mention the PowerKubeTM, then in line 75 you mention the StrikeMateTM. I realize that one has superseded the other (as is stated in lines 93-94), but as presented in lines 69 and 75+, it is very confusing to the reader. Please look at this.

Discussion

Many of the same issues arise here as was seen in the Introduction. You highlight that you did a “rigorous and systematic invertigation” then do not focus on the results for the reader.

For example, in line 200, you state your main purpose was to “perform a highly systematic investigation to determine the reliability of simulated impact strikes to the PowerKubeTM”. Then, in lines 204-206, you state that your hypothesis was confirmed “since the highest mean impact power resulted from level, centered strikes followed by level, off-center strikes, inclined center strikes, and inclined, off-center strikes.” That statement tells the reader nothing about the reliability of the PowerKubeTM (i.e., the stated purpose of the study). Please fix.

Lines 222 and 223 – Should StrikeKubeTM be the PowerKubeTM?

Round 2

Reviewer 1 Report

I disagree with the authors.It is necessary to perform the analyzes I have mentioned. 

They should add all the analyzes I wanted in previous revisions.

Author Response

√  We sincerely apologize to Reviewer 1 as we did not intend to be blunt on our previous Response to Reviewers.

√  We have included most of the previous recommendations (where appropriate) you have recommedned. This includes Bland-Altman plots (including 95% upper and lower confidence intervals, Figures 2 to 5), 95% CIs for the drop method (Table 1) and PowerKubeTM impact results (Power, Impact Power, Kinetic Energy and Power Index, Table 2). 

In subsection 3.4 we have included  Limits of Agreement (LoA), Standard Error of the Mean (SEM) and the Mean Absolute Percentage Error (MAPE) in the subsection 3.3.

However, with regard to Smallest Worthwhile Change (SWC) and minimal dectable change (MDC) we initially refer to the work by Hopkins (2004, SportsScience, 8, 1-7) and Edwards et al., (2022, Reliability and minimal detectable change of sprint times and force-velocity-power characteristics. The Journal of Strength & Conditioning Research36(1), 268-272) reported that  SWC and MDC are related to athletic performance where you assess baseline performance, then compute SWC or MDC to determine if the improvements in performance seen in post-testing were above the SWC or MDC, which would indicate a true physiological improvement. As our study was in vitro, with no human participants, our aim was not to assess human performance and then the change in performance, rather the reliability of the device. We believe the other outcome measures/analyses we have included recommended by Reviewer 1 demonstrate the reliability we were investigating.

We therefore request Reviewer 1 accept the additional analyses we have completed and amened our manuscript accordingly.